# Systematic Review of Mind–Body Modalities to Manage the Mental Health of Healthcare Workers during the COVID-19 Era

**DOI:** 10.3390/healthcare10061027

**Published:** 2022-06-01

**Authors:** Chan-Young Kwon, Boram Lee

**Affiliations:** 1Department of Oriental Neuropsychiatry, College of Korean Medicine, Dong-eui University, Busan 47227, Korea; 2Department of Clinical Korean Medicine, Graduate School, Kyung Hee University, Seoul 02453, Korea; qhfka9357@naver.com

**Keywords:** healthcare personnel, mental health, mind-body therapies, COVID-19, pandemics

## Abstract

Healthcare workers (HCWs) have suffered physical and psychological threats since the beginning of the coronavirus disease 2019 (COVID-19) pandemic. Mind-body modalities (MBMs) can reduce the long-term adverse health effects associated with COVID-specific chronic stress. This systematic review aims to investigate the role of MBMs in managing the mental health of HCWs during the COVID-19 pandemic. A comprehensive search was conducted using 6 electronic databases, resulting in 18 clinical studies from 2019 to September 2021. Meta-analysis showed that MBMs significantly improved the perceived stress of HCWs (standardized mean difference, −0.37; 95% confidence intervals, −0.53 to −0.21). In addition, some MBMs had significant positive effects on psychological trauma, burnout, insomnia, anxiety, depression, self-compassion, mindfulness, quality of life, resilience, and well-being, but not psychological trauma and self-efficacy of HCWs. This review provides data supporting the potential of some MBMs to improve the mental health of HCWs during COVID-19. However, owing to poor methodological quality and heterogeneity of interventions and outcomes of the included studies, further high-quality clinical trials are needed on this topic in the future.

## 1. Introduction

Globally, the coronavirus disease 2019 (COVID-19) pandemic has been a threat to mental and physical health of humanity [1]. Since the start of this pandemic, healthcare workers (HCWs) have suffered the physical and psychological threat of severe acute respiratory syndrome coronavirus 2 (SARS-CoV-2) [2,3]. In this context, the prevalence of anxiety, depression, and stress among HCWs in the COVID-19 pandemic was reported to be as high as 67.55%, 55.89%, and 62.99%, respectively [3]. Moreover, women, younger nurses, frontline HCWs, and workers in areas with higher infection rates are more likely to be severely affected by mental health effects in HCW during the pandemic [3]. Mental health difficulties of HCWs can lead to burnout, worsening attitudes toward patient safety, and hindering the efficient and safe use of medical resources when they are important [4]. As a result, several countries are implementing initiatives to improve health and well-being in HCW in the context of the COVID-19 pandemic, the most common of which are mental health initiatives [5].

Mind–body modality (MBM) can be defined as “*a health practice that combines mental focus, controlled breathing, and body movements to help relax the body and mind* [6]”. MBMs, including meditation, yoga, and mindfulness training, have been considered helpful in stress-related diseases by fostering resilience through self-care [7,8]. Researchers have also found that MBMs are effective in a variety of physical and psychological conditions, including chronic pain, anxiety, depression, cancer-related fatigue, tobacco addiction, inflammatory bowel disease, and cardiovascular disease [7]. MBMs are popular not only in patients with diagnosed diseases, but also in the general population. According to a survey in 2017, the use of yoga and meditation among American adults reached 14.3% and 14.2%, respectively [9]. Currently, MBMs are receiving attention as strategies to reduce the long-term adverse health effects associated with COVID-specific chronic stress and are being evaluated as specific, practical, affordable, and viable strategies to help manage chronic stress [10].

Maintaining the integrity of HCW’s mental health has emerged as an important topic in the COVID-19 pandemic [5], and the use of MBMs to manage and improve it and prevent progression to mental disorders has received attention [11]. In the context of the COVID-19 pandemic, some hospitals have introduced MBMs, including meditation and yoga, to protect the mental health of frontline HCWs [12,13]. Additionally, although contact-to-contact visits to psychiatric clinics have declined due to concerns about SARS-CoV-2 infection [14,15], MBMs are becoming more popular in combination with information and communications technology (ICT) [16].

As such, MBMs are likely to be a promising component of strategies for improving the mental health of HCW during the COVID-19 pandemic. However, how MBMs benefit the mental health aspects of HCWs in the context of COVID-19 has not yet been comprehensively reviewed. Therefore, this systematic review explores the roles of mind-body modalities in managing the mental health of HCWs during the COVID-19.

## 2. Materials and Methods

The protocol of this systematic review was published as a paper [17] and registered in the Open Science Framework registry (https://osf.io/tudbw, accessed on 13 August 2021). This systematic review complies with Preferred Reporting Items for Systematic Reviews and Meta-Analyses 2020 statement (Appendix A) [18].

### 2.1. Data Sources and Search Strategy

The following 6 electronic databases were searched for studies published from December 2019 (when the first case of COVID-19 was identified [19]) to September 2021: Medline (via PubMed), EMBASE (via Elsevier), Cochrane Central Register of Controlled Trials, Cumulative Index to Nursing and Allied Health Literature (via EBSCO), Allied and Complementary Medicine Database (via EBSCO), and PsycARTICLES (via ProQuest). In addition, we searched the reference lists of relevant articles and conducted a manual search on Google Scholar to include all of the relevant articles. We included both the literature published in peer-reviewed journals and gray literature, such as dissertations. We designed search strategies for all databases based on the advice of experts in the systematic review (Appendix A).

### 2.2. Eligibility Criteria

#### 2.2.1. Types of Study Design

Given the urgency of COVID-19, we included all types of original prospective quantitative intervention studies, including randomized controlled trials (RCTs), non-randomized controlled clinical trials (CCTs), and before-after studies. Retrospective and qualitative studies were excluded. There were no restrictions on publication language or publication status.

#### 2.2.2. Types of Participants

We included studies on all types of HCWs, such as physicians, nurses, hospital staff, and health managers, without restrictions on the sex, age, and ethnicity of the participants. However, we excluded studies that did not describe whether participants were directly or indirectly affected by COVID-19.

#### 2.2.3. Types of Interventions

As treatment interventions, MBMs, including meditation, mindfulness-based intervention, autogenic training, yoga, tai chi, qigong, breathing exercises, music therapy, guided imagery, biofeedback, prayer, and faith-based techniques were allowed. As control interventions, no treatment, waitlist, sham control, attention control, or active comparators were allowed.

#### 2.2.4. Types of Outcome Measures

The primary outcome was the level of perceived stress, assessed using validated tools, including the Perceived Stress Scale [20]. Secondary outcomes included mental health-related outcomes, such as depression, anxiety, burnout, and safety data of the intervention.

### 2.3. Study Selection

A study selection was conducted through a three-step screening process based on the eligibility criteria. First, titles and/or abstracts of the searched studies were screened to identify potentially eligible articles. Second, potentially eligible reports were sought for retrieval. Third, the full text of the retrieved reports was reviewed. Two researchers (C.-Y.K. and B.L.) independently conducted the study selection process, and disagreements between the researchers were resolved through discussion.

### 2.4. Data Extraction

Two researchers (C.-Y.K. and B.L.) independently extracted the following information using a predefined, pilot-tested excel form: the first author’s name, year of publication, country, study design, sample size, details of participants, treatment and control interventions, a treatment period of intervention, outcome measures, results, and safety data. Any discrepancies between the researchers were resolved through discussion. When the data were insufficient or ambiguous, the corresponding authors of the original studies were contacted via e-mail.

### 2.5. Methodological Quality and Risk of Bias Assessment

The methodological quality of the included studies was assessed using the corresponding critical appraisal skills program tools, depending on the study type [21]. For RCTs, the Cochrane Collaboration risk of bias tool was used to assess the related risk of bias [22]. For CCTs and before-after studies, the Quality Assessment of Controlled Intervention Studies and the Quality Assessment Tool for Before-After (Pre-Post) studies with no control group by the National Heart, Lung, and Blood Institute were used to assess methodological quality [23]. Two researchers (C.-Y.K. and B.L.) independently assessed the methodological quality and risk of bias of the included studies and any disagreements between the researchers were resolved through discussion.

### 2.6. Data Analysis and Synthesis

Descriptive analyses of the participants, interventions, controls, and outcomes of all of the studies were performed. If there were 2 or more RCTs or CCTs with the same outcome measures, a meta-analysis was performed using RevMan 5.4 (the Cochrane Collaboration, London, UK). In the meta-analysis, dichotomous and continuous data were presented as risk ratios (RR) and standardized mean differences (SMD) with 95% confidence intervals (CIs). The *I*^2^ values of ≥50% and ≥75% were considered substantial and statistically heterogeneous, respectively. In the meta-analysis, a random-effects model was used if the included studies had significant heterogeneity (*I*^2^ value ≥ 50%), whereas the fixed-effect model was used when the heterogeneity was insignificant or the number of studies included in the meta-analysis was less than five [24]. If sufficient data were available, subgroup analyses were planned according to the (a) type of HCWs and (b) type of mind-body modality. In addition, sensitivity analyses were conducted to identify the robustness of the results by excluding (a) studies with a high risk of bias and (b) data outliers. Evidence of publication bias was assessed using funnel plots if at least ten RCTs were included in each meta-analysis. The results of the included before-and-after studies were only described without quantitative synthesis.

## 3. Results

### 3.1. Study Search

In the initial search, 2816 documents were found, and 263 duplications were removed. Among the 2553 documents, 46 potentially relevant articles were selected after title and abstract screening. On the full-text screening, fifteen non-clinical studies [25,26,27,28,29,30,31,32,33,34,35,36,37,38,39], four non-intervention studies [40,41,42,43], three generic studies [44,45,46], five studies not using MBMs [47,48,49,50,51], and one not presenting evaluation results [52] were excluded. Ultimately, this review included 18 studies [53,54,55,56,57,58,59,60,61,62,63,64,65,66,67,68,69,70] (Figure 1).

### 3.2. Characteristics of Included Studies

Among the included studies, 16 [53,54,56,57,58,59,60,61,62,63,64,65,66,68,69,70] were published articles and 2 [55,67] were conference abstracts. Researchers from nine countries conducted these studies, and the study conducted in the United States was the most common, with six studies [54,55,59,60,63,68], followed by India with three studies [56,57,69]. Among the included studies, five were RCTs [58,65,66,68,69], four were CCTs [57,59,62,70], and the remaining ten were before-after studies [53,54,55,56,60,61,63,64,67]. Except for one study [69], the types of HCWs in the included subjects were all described. Among them, 14 [53,54,56,57,58,59,60,61,63,64,65,66,68,70] included nurses (14/17, 82.35%), 9 [53,54,55,56,58,62,63,67,68] included clinicians (9/17, 52.94%), and 6 [54,56,58,60,63,68] included non-clinical workers (6/17, 35.29%). The sample sizes of the included studies ranged from 3 to 482, with a mean of 102.44. Although the MBMs used in the included studies were heterogeneous, their components could be classified into nine types: music, mindfulness, wellness/welfare/well-being, yoga, meditation, empathy/self-compassion, breathing exercise, relaxation, and guided imagery. The interventions included an average of 1.61 mind-body modality components, with mindfulness being the most common component (6/18, 33.33%), followed by breathing exercises (5/18, 27.78%), yoga (4/18, 22.22%), wellness/welfare/well-being, meditation, and relaxation (3/18, 16.67%), music and empathy/self-compassion (2/18, 11.11%, respectively), and guided imagery (1/18, 5.56%). Except for 2 studies that did not describe the duration of intervention [55,57], the duration varied from 1 to 150 days, with a mean of 36.75 days (Table 1).

### 3.3. Methodological Qualities of Included Studies

#### 3.3.1. RCTs

Three RCTs [58,65,68] described appropriate random sequence generation methods such as computerized randomization; conversely, the other two studies [66,69] did not describe this method. No study has described the method of allocation concealment. Fiol-DeRoque et al. [58] conducted a study using a mobile application and reported that double-blind was implemented. The remaining four RCTs [65,66,68,69] did not report the implementation of blinding but were evaluated as high due to the nature of the intervention. Only one study [58] reported that blinding of the outcome assessment was not performed; in the remaining studies [65,66,68,69], blinding of the outcome assessor was not described. For incomplete outcome data, one study [58] performed an intention-to-treat analysis, and another study [66] with no dropouts was rated low in this domain. In the remaining three studies [65,68,69], dropouts existed, but the cause was not described, and a per-protocol (PP) analysis was performed; therefore, this domain was rated highly. With regard to selective reporting, the protocol was confirmed in only one study [58], and the pre-planned outcome was confirmed to be reported and evaluated as low. Other studies [65,66,68,69] evaluated selective reporting as unclear. Three studies [65,66,68], in which clinical and demographic homogeneity between the two groups was confirmed at baseline, were evaluated as low in other sources of bias, and the remaining studies were evaluated as unclear (Appendix A).

#### 3.3.2. CCTs

As the included CCTs were not RCTs, they were evaluated as “no” in Q1 and “not applicable (NA)” in Q2. Allocation concealment and blinding procedures were not reported in any of the studies. Therefore, Q3 and Q5 were evaluated as not reported (NR), and Q4 was evaluated as “no,” considering that double-blinding was impossible due to the nature of the intervention. In a study [59], statistical heterogeneity of baseline characteristics was reported, and it was evaluated as “no” in Q6; the heterogeneity was not described in the remaining studies [57,62,70]. In another study [62], the overall dropout rate from the study at the endpoint was more than 20% of the number allocated to treatment, and the differential dropout rate at the endpoint was more than 15 percent. Therefore, Q7 and Q8 of this study [62] were evaluated as “no.” As treatment adherence was not reported in all of the studies, Q9 was evaluated as NR. As only one study [57] recommended avoidance of other interventions in the groups, it was evaluated as “yes” in Q10. As validated outcomes were used in all of the studies, they were evaluated as “yes” in Q11. The sample size was calculated in only one study [70] and was evaluated as “yes” in Q12. The pre-registered protocol was confirmed in only one study [62], its Q13 was evaluated as “yes,” and the remaining studies [57,59,70] were evaluated as cannot be determined. Two studies [57,70] with no dropouts were evaluated as NA in Q14, whereas the other two studies [59,62] with PP analysis were evaluated as “no” (Appendix A).

#### 3.3.3. Before-after Studies

The purpose of the studies was clearly described; therefore, it was evaluated as “yes” in Q1. As the selection criteria for the study population were not clearly described in the four studies [54,55,56,60], Q2 and CD were not evaluated in Q4. Except for one study [61] that included only geriatric fellows as participants, other studies [53,54,55,56,60,63,64,67] included two or more occupations, so their Q3 were evaluated as “yes.” In only one study [61], the sample size was calculated and evaluated as “yes” in Q5. Because the intervention was insufficiently described in two studies [55,67], it was evaluated as “no” in their Q6. Two studies [53,63] that did not use a validated outcome were rated as “no” in their Q7. Only one study [64] reported that the outcome assessor was blinded. As not described in other studies [53,54,55,56,60,61,63,67], all were assessed as CD in their Q8. In two studies [54,60], loss to follow-up after baseline was >20%; therefore, it was evaluated as “no” in their Q9. One study [63] without a statistical test for pre-to post-changes, was rated as “no” in Q10. In only one study [56], the outcome assessment was performed three times, and it was evaluated as “yes” in Q11. In other studies [53,54,55,60,61,63,64,67], it was evaluated as “no” in Q11 because it was only evaluated twice: before and after. There was no intervention at the group level; therefore, all were evaluated as NA in Q12 (Appendix A).

### 3.4. Main Results

The outcomes used among the included studies varied, but they can be classified into 12 categories: perceived stress (the primary outcome), psychological trauma, burnout, insomnia, self-efficacy, anxiety, depression, self-compassion, mindfulness, and quality of life (QOL), resilience, and well-being. Among these, meta-analyses of stress, depression, and anxiety are possible.

In terms of perceived stress, a meta-analysis of Depression, Anxiety and Stress Scale (DASS), including two RCTs and one CCT, was possible. Consequently, MBMs showed a significantly greater effect on improving perceived stress compared to the control interventions (SMD, −0.37; 95% CI, −0.53 to −0.21; *I*^2^ = 96%). However, in the individual analysis, compared to no intervention, yoga and music-based intervention had a significant effect on perceived stress improvement (SMD, −1.88; 95% CI, −2.35 to −1.41), whereas self-compassion-based intervention had no significant effect (SMD, −0.09; 95% CI, −0.66 to 0.47). Otherwise, cognitive-behavioral therapy (CBT) and mindfulness-based intervention showed borderline significance compared to the psychoeducational intervention (SMD, −0.18; 95% CI, −0.36 to −0.00) (Figure 2a). Regarding depression, a meta-analysis of DASS, including two RCTs and one CCT, was possible. Consequently, MBMs showed a significantly greater effect on improving depression than the control interventions (SMD, −0.29; 95% CI, −0.45 to −0.12; *I*^2^ = 98%). However, in the case of individual analysis, only yoga- and music-based intervention had a significant effect on depression improvement compared to no intervention (SMD, −2.82; 95% CI, −3.37 to −2.26). Self-compassion-based intervention and CBT and mindfulness-based intervention, on the other hand, did not show significant effect compared to control interventions (SMD, −0.26; 95% CI, −0.83 to 0.31; SMD, −0.02; 95% CI, −0.20 to 0.16) (Figure 2b). Regarding anxiety, a meta-analysis of DASS, including two RCTs and one CCT, was possible. Consequently, MBMs showed a significantly greater effect on improving anxiety compared to the control interventions (SMD, −0.43; 95% CI, −0.59 to −0.27; *I*^2^ = 96%). However, in the case of individual analysis, compared to no intervention, yoga and music-based intervention had a significant effect on anxiety improvement (SMD, −2.21%; CI, −2.71 to −1.72), whereas self-compassion-based intervention had no significant effect (SMD, −0.22; 95% CI, −0.79 to 0.35). CBT and mindfulness-based intervention also significantly improved anxiety compared to psychoeducational intervention (SMD, −0.22; 95% CI, −0.39 to −0.04) (Figure 2c). MBMs for each outcome showed a significantly positive effect compared to the control group or before treatment as follows (in descending order): self-compassion (2/2, 100%), QOL (2/2, 100%), perceived stress (6/8, 75%), resilience (4/6, 66.67%), burnout (3/5, 60%), insomnia (3/5, 60%), well-being (1/2, 50%), anxiety (3/7, 42.86%), depression (3/8, 37.5%), mindfulness (1/4, 25%), psychological trauma (0/1, 0%), and self-efficacy (0/1, 0%) (Table 2).

### 3.5. Safety Data

None of the included studies reported adverse events or safety data.

### 3.6. Publication Bias

As fewer than ten studies were included in the meta-analysis, publication bias through funnel plot generation was not evaluated.

## 4. Discussion

### 4.1. Main Findings

This review was performed in order to investigate the benefits of MBMs on the mental health aspects of HCWs in the context of COVID-19. Through a comprehensive literature search, 18 studies [53,54,55,56,57,58,59,60,61,62,63,64,65,66,67,68,69,70] were included. According to the meta-analysis, MBMs had a significantly positive effect on the perceived stress of HCWs, which was the primary outcome of this study. Regarding the types of individual MBMs, the effects of yoga- and music-based interventions appeared to be the most prominent. In a meta-analysis of depression and anxiety, MBMs showed a significantly positive improvement compared to the control group. In this case, yoga- and music-based interventions had the largest effect size. However, the effect of self-compassion-based interventions on stress, depression, and anxiety was not significant. In other words, the effects of MBMs on mental health of HCWs may differ according to individual MBMs. For individual outcomes, some MBMs had significant positive effects on psychological trauma, burnout, insomnia, anxiety, depression, self-compassion, mindfulness, QOL, resilience, and well-being, but not psychological trauma and self-efficacy, compared to controls (in RCTs and CCTs) or baseline (in before-after studies). Although these results provide data that some MBMs may be useful options for mental health management of HCWs in the context of COVID-19, the methodological quality of the included studies was not optimal. In addition, the number of RCTs performed with rigorous design was insufficient, and CCT or before-after studies accounted for more than half of the studies. Therefore, the findings of this study could be greatly influenced by the results of large-scale rigorous clinical studies in the future.

### 4.2. Clinical Implications

The mental health of HCWs in the context of COVID-19 is a serious threat [3,4], and measures to manage it are urgently needed [5]. At present, to protect the mental health of this population and reduce stress as much as possible, there is an emphasis on establishing tailored, effective stress reduction interventions [71]. To develop effective anti-stress interventions, empirical evidence exists for some MBMs, such as mindfulness-based interventions, diaphragmatic respiration, and acting on self-efficacy [71]. Breath-focused mind-body therapies, as a strategy in precision medicine, have recently been claimed by some researchers to be effective in managing stress and anxiety [72,73]. Therefore, leveraging MBMs to manage the mental health of HCWs in the context of COVID-19 could be a promising strategy [10].

According to the results of the studies included in this review, some MBMs have shown positive effects in improving the mental health aspects of HCWs. When classified as individual outcomes, MBMs showed a relatively high rate of positive effects on self-compassion, QOL, perceived stress, resilience, burnout, insomnia, and well-being in this population. In addition, the individual MBMs used were heterogeneous, and no studies have compared two or more MBMs in this population. Therefore, it was difficult to find the optimal MBMs for mental health and psychological stress management in HCWs during the COVID-19 era. Nevertheless, according to the meta-analysis of the review, yoga- and music-based interventions had a larger effect size on perceived stress (DASS) than CBT and mindfulness-based or self-compassion-based interventions. A recent meta-analysis also found that yoga may help relieve stress in people who live under high stress or negative emotions, including HCWs, and the mechanism may be related to the modulation of sympathetic-vagal balance [74,75]. Moreover, in a systematic review of MBMs for nurses, yoga may be helpful in improving burnout and perceived stress among nurses in hospital setting [76].

Important considerations should be taken into account when introducing interventions for mental health management of HCWs in the context of COVID-19. The reality of a busy clinical setting must be considered. Strategies for managing stress and mental health in this population should be feasible and accessible. In this context, more than half of the included studies [53,54,56,58,60,61,62,63,65,68,69,70] (12/18, 66.67%) provided participants with MBMs combined with ICT, including mobile phone applications, guided audio files, video files, and video conferences. In addition, three studies [55,64,67] integrated MBMs into the work environment of participants, such as the educational curriculum and time for a shift. These results suggest that MBMs may be introduced to reflect the work environment of HCWs in the context of COVID-19.

### 4.3. Limitations and Suggestions for Further Studies

This systematic review has several limitations. First, the MBMs and outcomes used in the included studies were heterogeneous. Therefore, quantitative synthesis was not possible for most outcomes in this study. However, given that the mental health of HCWs in the context of COVID-19 has important clinical relevance and MBMs are a promising option, future research in this field needs to be further standardized and refined. In particular, a head-to-head trial may be attempted to investigate the most effective MBMs for improving perceived stress among HCWs during the COVID-19 era. Second, the methodological quality of the included studies was suboptimal. These methodological limitations may negatively affect the reliability of this study’s findings. In addition, it was rare among the included studies to be conducted with strict designs, including RCT (5/18, 27.78%). Future research in this field will be able to reflect a research design that reflects the characteristics of MBMs. For example, to investigate the effectiveness and safety of MBMs, n-of-1 trials may be ideal than traditional RCT [77]. Third, this study did not consider the temporal and environmental effects of COVID-19. Given that the impact of COVID-19 on the mental health of HCWs may vary depending on the time, place, and clinical setting, future research in this field will be able to develop customized interventions that take into account the temporal and environmental impacts of COVID-19 on HCWs.

## 5. Conclusions

This is the most comprehensive review available on the impact of MBMs on the mental health of HCWs during the COVID-19 era. This review provides data supporting the potential of some MBMs to improve the mental health of HCWs during COVID-19. There is evidence that yoga- and music-based interventions are helpful for improvement for perceived stress, the primary outcome. In addition, some MBMs had significant positive effects on psychological trauma, burnout, insomnia, anxiety, depression, self-compassion, mindfulness, QOL, resilience, and well-being, but not psychological trauma and self-efficacy of HCWs. However, owing to poor methodological quality and heterogeneity of interventions and outcomes of the included studies, further high-quality clinical trials are needed on this topic in the future.

## Figures and Tables

**Figure 1 healthcare-10-01027-f001:**
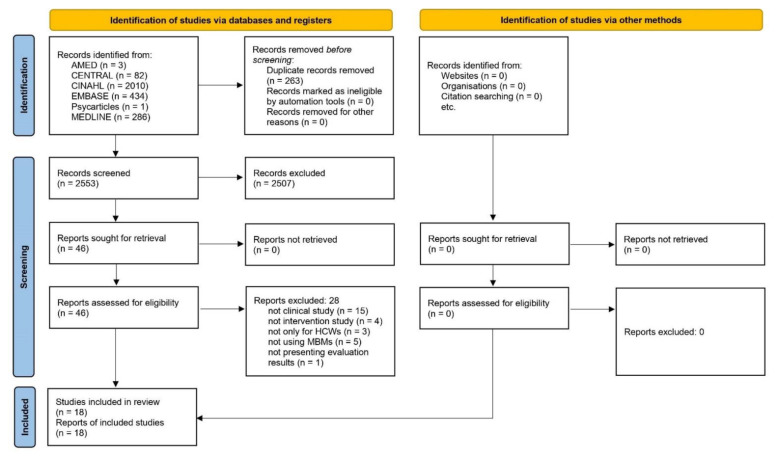
PRISMA flow chart of this review.

**Figure 2 healthcare-10-01027-f002:**
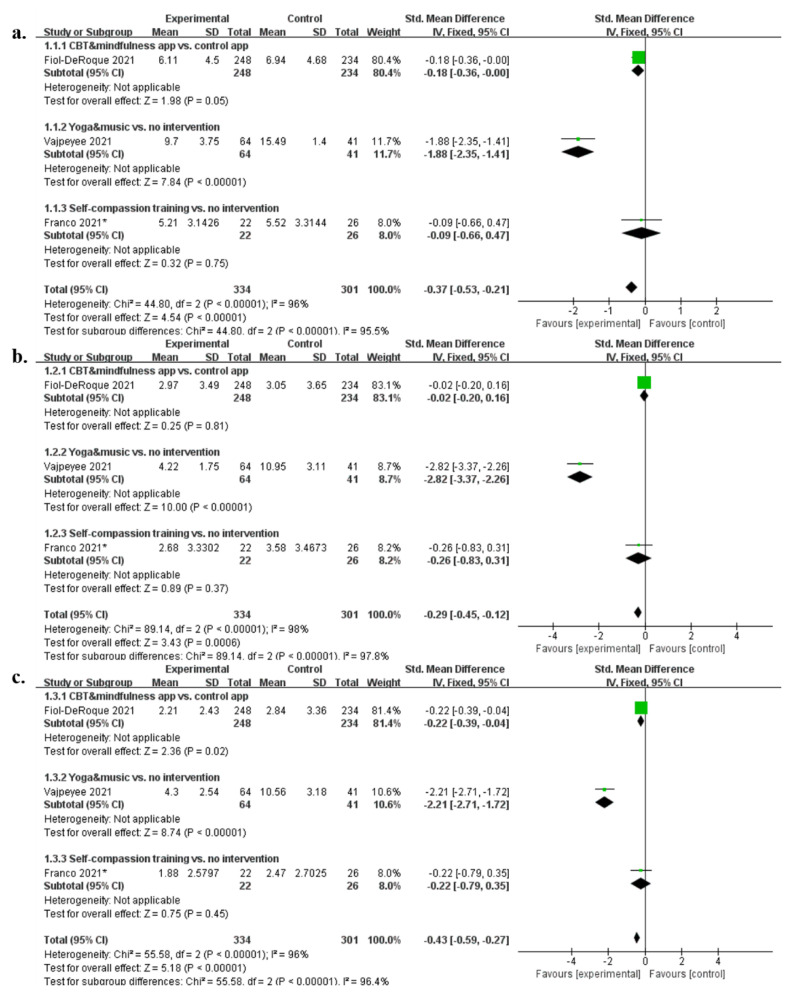
Meta-analysis results of Depression, Anxiety and Stress Scale: (**a**) stress; (**b**) depression; and (**c**) anxiety. Note. *, non-randomized controlled clinical trials.

**Table 1 healthcare-10-01027-t001:** Characteristics of included studies.

Author (Country)	Population	Sample Size (Included→Analyzed)	Intervention (Delivery)	Treatment Period	Outcomes
RCTs (*n* = 5)					
Fiol-DeRoque 2021 [58] (Spain)	HCWs (including physicians, nurses, and others)	EG: 248→248 CG: 234→234 (ITT analysis)	Self-managed psychoeducational intervention, based on CBT and mindfulness vs. Control app (Mobile phone-based intervention)	14 days	1. DASS-21; 2. DTS; 3. MBI–HSS; 4. ISI; 5. General Self-Efficacy Scale
Nourian 2021 [65] (Iran)	Nurses working in COVID-19 care wards	44→41 *	MBSR vs. waitlist (WhatsApp group, otherwise unclear)	7 weeks	1. PSQI
Sanadgol 2021 [66] (Iran)	ICU nurses	EG: 25→25 CG: 25→25	Guided imagery training vs. no intervention (1. 90 min of the group training session (two sessions); 2. at least three sessions a week at home)	1 month	1. 15-item DAS
Thimmapuram 2021 [68] (US)	Physicians and advance practice providers (including physicians, nurse, and other hospital staff)	EG: 77→41 ^+^ CG: 78→58 ^+^	Heartfulness meditation vs. usual clinical care (Home practice: daily 6 min of heartfulness meditation via audio file)	4 weeks	1. UCLA Loneliness Scale Version 3; 2. PSQI
Vajpeyee 2021 [69] (India)	HCWs (the types of HCWs were not described)	240→209 (EG: 116, CG: 93) ^+^	Yoga and music intervention vs. no intervention (1. Daily 30 min of yoga and music by WhatsApp video; 2. Help of social workers to motivate the participants by personal visits or telephonic communication)	30 days	1. DASS-42
CCTs (*n* = 4)					
Emmanuel 2021 [57] (India)	ICU nurses	EG: 30→30 CG: 30→30	Welfare program including stress management and breathing exercise vs. NR (30 min for breathing exercise, otherwise unclear)	NR	1. ENSS
Franco 2021 [59] (US)	Pediatric nurses	53→48 (EG: 22, CG: 26) ^+^	Self-compassion training program vs. no intervention (One day training program, 6 h)	1 day	2 weeks post-intervention 1. SCS; 2. CAMS; 3. Compassion Scale; 4. ProQOL; 5. DASS; 6. Resiliency activation and decompression and job engagement
Luton 2021 [62] (UK)	Surgical trainees in a single UK statutory education body	EG: 24→14 ^+^ CG: 14→14	Enhanced stress-resilience training course (mindfulness-based exercises) vs. no intervention (1. Weekly 75 min online tutorials (15 min of debrief, and 60 min of training course); 2. Daily up to 20 min of mindfulness exercises following guided media)	5 weeks	1. MBI; 2. PSS; 3. PHQ-2; 4. CAMS–R; 5. STAI
Ibrahim 2022 [70] (Indonesia)	Nurses who worked in COVID-19 patient services of health care facilities	EG: 25→25 CG: 25→25	Mindfulness breathing meditation vs. waitlist (Twice a week, 15 min (video practice) per session in WhatsApp group)	4 weeks	1. WEMWBS Indonesian version
Before-after study (*n* = 9)					
Giordano 2020 [53] (Italy)	Clinical staff in COVID-19 unit (including physicians and nurses)	34→29 ^+^	Tailored music therapy (15–20 min, mobile phones, otherwise unclear)	4 weeks	Before listening to music, and within one hour after the end of listening 1. MusicTeamCare-Q1 End of the study 2. MusicTeamCare-Q2
Klatt 2020 [54] (US)	HCWs (including physicians, nurses, other clinical staff, and non-clinical healthcare staff)	465→267 ^+^	Mindfulness-based intervention (Mindfulness in Motion) (1. 12 min, weekly group meeting; 2. Individual practice by using audio or video practice (via smartphone or computer))	8 weeks	1. MBI; 2. PSS; 3. CDRS; 4. UWES
Coffey 2021 [55] (US)	Geriatric medicine fellows	3	Wellness program (Wellness, Empathy, and Philanthropy) (Incorporate the wellness program to curriculum)	NR	1. Abbreviated MBI; 2. Brief Resilience Scale
Divya 2021 [56] (India)	HCWs (including physicians, nurses, other clinical staff, and non-clinical healthcare staff)	100→92 ^+^	Yogic breathing technique (1. 4-day online breath and meditation workshop (video conference); 2. 35 min home practice)	40 days	1. DASS-21; 2. PSQI; 3. CDRS; 4. SWLS
Heeter 2021 [60] (US)	Hospice HCWs (including nurses, other clinical staff, and non-clinical healthcare staff)	151→76 ^+^	Yoga-based meditation (1. Half-hour online introduction to the program; 2. Weekly meditation session at each team’s monthly all-staff meeting (12 min, accessible via the program website))	6 weeks	1. Brief PFI; 2. MAIA
Liu 2021 [61] (China)	Clinical first-line nurses in COVID-19 designated hospitals	151→140 ^+^	Diaphragmatic breathing relaxation training (Daily, 30 min at 8 PM via MP3 audio and demonstration video)	4 weeks	1. PSQI; 2. SAS; 3. SDS
Narayanan 2021 [63] (US)	HCWs (including physicians, nurses, other clinical staff, and non-clinical healthcare staff)	100→88 ^+^	Breathing practice and meditation (Simha kriya) (5 min, one to two times daily, Video with instruction)	4 weeks	1. Meditation perception questionnaire
Nijland 2021 [64] (Netherlands)	ICU nurses in an academic hospital	86	VR relaxation (high-quality immersive 360-degree videos of calming natural environments) (10 min during their shift, otherwise unclear)	1 session	1. PSS; 2. CDRS
So 2021 [67] (UK)	Junior doctors working at a single UK cancer center	10	Well-being program including breathing and relaxation exercise, clinical debriefing, reflective practice, and mindfulness strategies (Weekly 30 min well-being sessions throughout their oncology placement)	4–6 months	1. WEMWBS

Abbreviations. CAMS–R, Cognitive and Affective Mindfulness Scale–Revised; CBT, cognitive-behavioral therapy; CCT, non-randomized controlled clinical trial; CDRS, Connor–Davidson Resilience Scale; CG, control group; COVID–19, Coronavirus Disease 2019; DAS, Templer Death Anxiety Scale; DASS, Depression, Anxiety and Stress Scale; DTS, Davidson Trauma Scale; EG, experimental group; ENSS, Expanded Nursing Stress Scale; HCW, healthcare worker; ICU, intensive care unit; ISI, Insomnia Severity Index; ITT, intent–to–treat; MAIA, Multidimensional Assessment of Interoceptive Awareness; MBI–HSS, Maslach Burnout Inventory–Human Services Survey; MBI, Maslach Burnout Inventory; MBSR, mindfulness-based stress reduction; NR, not reported; PFI, Stanford Professional Fulfillment Index; PHQ, Patient Health Questionnaire; PP, per-protocol; ProQOL, Professional Quality of Life; PSQI, Pittsburgh sleep quality index; PSS, Perceived Stress Scale; RCT, randomized controlled clinical trial; SAS, Self–rating Anxiety Scale; SCS, Self-Compassion Scale; SDS, Self–rating Depression Scale; STAI, State–Trait Anxiety Inventory; SWLS, Satisfaction With Life Scale; UWES, Utrecht Work Engagement Scale; VR, virtual reality; WEMWBS, Warwick–Edinburgh Mental Well-being Scale. Note. *, The numbers of participants of EG and CG were unclear; ^+^, The reason for the dropout is not stated.

**Table 2 healthcare-10-01027-t002:** Main results of included studies.

Outcomes	Comparison (Treatment Period)	Results	Reference
Outcomes related to perceived stress			
1. DASS (perceived stress)	CBT and mindfulness-based app vs. Psychoeducation app (14 days)	NS (*p* > 0.05)	Fiol-DeRoque 2021 [58]
	**Yoga and music intervention** vs. no intervention (30 days)	Participants with baseline abnormality: EG < CG (*p* = 8.28 × 10^−19^) Participants without baseline abnormality: No statistical comparison between groups	Vajpeyee 2021 [69]
	**Self-compassion training** vs. no intervention (1 day)	EG < CG (*p* < 0.05)	Franco 2021 [59]
	Yogic breathing technique (40 days)	NS (*p* = 0.49)	Divya 2021 [56]
2. ENSS (nursing stress)	**Welfare program including breathing exercise** vs. no intervention (NR)	EG: pre < post (*p* < 0.05) CG: NS (*p* > 0.05)	Emmanuel 2021 [57]
3. PSS (perceived stress)	**Mindfulness-based stress-resilience training** vs. no intervention (5 weeks)	EG < CG (*p* < 0.01)	Luton 2021 [62]
	**Mindfulness-based movement** (8 weeks)	pre > post (*p* = 0.00001)	Klatt 2020 [54]
	**VR relaxation** (1 session)	pre > post (*p* < 0.005)	Nijland 2021 [64]
Outcomes related to psychological trauma			
1. DTS (psychological trauma)	CBT and mindfulness-based app vs. Psychoeducation app (14 days)	NS (*p* > 0.05)	Fiol-DeRoque 2021 [58]
Outcomes related to burnout			
1. MBI–HSS (burnout)	CBT and mindfulness-based app vs. Psychoeducation app (14 days)	(1) Emotional exhaustion: NS (*p* > 0.05); (2) Professional accomplishment: NS (*p* > 0.05) (3) depersonalization: NS (*p* > 0.05)	Fiol-DeRoque 2021 [58]
2. MBI (burnout)	Mindfulness-based stress-resilience training vs. no intervention (5 weeks)	NS (*p* = 0.630)	Luton 2021 [62]
	**Mindfulness-based movement** (8 weeks)	(1) Total: pre > post (*p* = 0.00001); (2) Emotional exhaustion: pre > post (*p* = 0.00001); (3) Depersonalization: pre > post (*p* = 0.0012); (4) Personal accomplishment: pre > post (*p* = 0.00001)	Klatt 2020 [54]
3. Brief MBI (burnout)	Wellness program (NR)	Statistical analysis was not performed.	Coffey 2021 [55]
4. Brief PFI (burnout)	**Yoga-based meditation** (6 weeks)	(1) Professional fulfillment: pre > post (*p* = 0.008); (2) Work exhaustion: pre > post (*p* = 0.049); (3) Interpersonal disengagement: NS (*p* > 0.05); (4) Total: NS (*p* > 0.05)	Heeter 2021 [60]
Outcomes related to insomnia			
1. ISI (insomnia severity)	CBT and mindfulness-based app vs. Psychoeducation app (14 days)	NS (*p* > 0.05)	Fiol-DeRoque 2021 [58]
2. PSQI (sleep quality)	**MBSR** vs. no intervention (7 weeks)	(1) Global score: NS (*p* = 0.105); (2) Subjective sleep quality: EG < CG (*p* = 0.000); (3) Sleep latency: EG < CG (*p* = 0.020); (4) Sleep duration: NS (*p* = 0.084); (5) Habitual sleep efficiency: NS (*p* = 0.148); (6) Sleep disturbances: NS (*p* = 0.587); (7) Use of sleep medication: NS (*p* = 0.118); (8) Daytime drowsiness: NS (*p* = 0.050)	Nourian 2021 [65]
	**Heartfulness meditation** vs. usual care (4 weeks)	EG: pre > post (*p* = 0.001) CG: NS (*p* = 0.122)	Thimmapuram 2021 [68]
	Yogic breathing technique (40 days)	NS (*p* = 0.154)	Divya 2021 [56]
	**Diaphragmatic breathing relaxation training** (4 weeks)	(1) Global: pre > post (*p* < 0.001); (2) Subjective sleep quality: pre > post (*p* < 0.001); (3) Sleep duration: pre > post (*p* < 0.001); (4) Sleep latency: pre > post (*p* < 0.001); (5) Habitual sleep efficiency: pre > post (*p* = 0.015); (6) Sleep disturbances: pre > post (*p* < 0.001); (7) Use of sleeping medication: NS (*p* = 0.134); (8) Daytime dysfunction: pre > post (*p* = 0.001)	Liu 2021 [61]
Outcomes related to self-efficacy			
1. General Self-Efficacy Scale (self-efficacy)	CBT and mindfulness-based app vs. Psychoeducation app (14 days)	NS (*p* > 0.05)	Fiol-DeRoque 2021 [58]
Outcomes related to anxiety			
1. DASS (anxiety)	CBT and mindfulness-based app vs. Psychoeducation app (14 days)	NS (*p* > 0.05)	Fiol-DeRoque 2021 [58]
	**Yoga and music intervention** vs. no intervention (30 days)	Participants with baseline abnormality: EG < CG (*p* = 1.02 × 10^−16^) Participants without baseline abnormality: No statistical comparison between groups	Vajpeyee 2021 [69]
	Self-compassion training vs. no intervention (1 day)	NS (*p* = 0.05)	Franco 2021 [59]
	Yogic breathing technique (40 days)	NS (*p* = 0.613)	Divya 2021 [56]
2. 15-item DAS (death anxiety)	**Guided imagery training** vs. no intervention (1 month)	EG < CG (*p* = 0.004)	Sanadgol 2021 [66]
3. STAI (anxiety)	Mindfulness-based stress-resilience training vs. no intervention (5 weeks)	NS (*p* = 0.450)	Luton 2021 [62]
4. SAS (anxiety)	**Diaphragmatic breathing relaxation training** (4 weeks)	pre > post (*p* < 0.001)	Liu 2021 [61]
Outcomes related to depression			
1. DASS (depression)	CBT and mindfulness-based app vs. Psychoeducation app (14 days)	NS (*p* > 0.05)	Fiol-DeRoque 2021 [58]
	**Yoga and music intervention** vs. no intervention (30 days)	Participants with baseline abnormality: EG < CG (*p* = 2.55 × 10^−18^) Participants without baseline abnormality: No statistical comparison between groups	Vajpeyee 2021 [69]
	Self-compassion training vs. no intervention (1 day)	NS (*p* = 0.23)	Franco 2021 [59]
	Yogic breathing technique (40 days)	NS (*p* = 0.563)	Divya 2021 [56]
2. UCLA Loneliness Scale Version 3 (loneliness)	**Heartfulness meditation** vs. usual care (4 weeks)	EG: pre > post (*p* = 0.009) CG: NS (*p* = 0.254)	Thimmapuram 2021 [68]
3. PHQ-2 (depression)	Mindfulness-based stress-resilience training vs. no intervention (5 weeks)	NS (*p* > 0.05)	Luton 2021 [62]
4. MusicTeamCare-Q1 (tiredness, sadness, fear, and worry)	**Tailored music therapy** (4 weeks)	(1) Breathing playlist: the scores of four symptoms were all significantly decreased (*p* < 0.05); (2) Energy playlist: the scores of four symptoms were all significantly decreased (*p* < 0.05); (3) Breathing customized playlist: the scores of sadness, fear, and worry were significantly decreased (*p* < 0.05); (4) Energy customized playlist: the scores of four symptoms were all significantly decreased (*p* < 0.05); (5) Serenity customized playlist: the scores of sadness, fear, and worry were significantly decreased (*p* < 0.05).	Giordano 2020 [53]
5. SDS (depression)	Diaphragmatic breathing relaxation training (4 weeks)	NS (*p* = 0.359)	Liu 2021 [61]
Outcomes related to self-compassion			
1. SCS (self-compassion)	**Self-compassion training** vs. no intervention (1 day)	EG > CG (*p* < 0.001)	Franco 2021 [59]
2. Compassion scale (compassion)	**Self-compassion training** vs. no intervention (1 day)	EG < CG (*p* < 0.01)	Franco 2021 [59]
Outcomes related to mindfulness			
1. CAMS (mindfulness)	**Self-compassion training** vs. no intervention (1 day)	EG > CG (*p* < 0.001)	Franco 2021 [59]
	Mindfulness-based stress-resilience training vs. no intervention (5 weeks)	NS (*p* > 0.05)	Luton 2021 [62]
2. MAIA (interoceptive awareness)	Yoga-based meditation (6 weeks)	(1) Self-regulation: NS (*p* > 0.05); (2) Attention regulation: NS (*p* > 0.05); (3) Emotional awareness: NS (*p* > 0.05); (4) Body noticing: NS (*p* > 0.05); (5) Body listening: NS (*p* > 0.05); (6) Body trusting: NS (*p* > 0.05); (7) Total: NS (*p* > 0.05)	Heeter 2021 [60]
3. Meditation Perception Questionnaire (perception about meditation)	Breathing practice and meditation (4 weeks)	Statistical analysis was not performed.	Narayanan 2021 [63]
Outcomes related to QOL			
1. ProQOL (professional QOL)	**Self-compassion training** vs. no intervention (1 day)	(1) Compassion satisfaction: EG > CG (*p* = 0.01); (2) Burnout: EG < CG (*p* < 0.01); (3) Secondary traumatic stress: NS (*p* = 0.08)	Franco 2021 [59]
2. SWLS (satisfaction with life)	**Yogic breathing technique** (40 days)	pre < post (*p* < 0.001)	Divya 2021 [56]
Outcomes related to resilience			
1. Resiliency activation and decompression and job engagement (resilience and job engagement)	**Self-compassion training** vs. no intervention (1 day)	(1) Resiliency activation: NS (*p* = 0.55); (2) Resiliency decompression: EG < CG (*p* < 0.01); (3) Job engagement: NS (*p* = 0.21)	Franco 2021 [59]
2. CDRS (resilience)	**Mindfulness-based movement** (8 weeks)	pre < post (*p* = 0.00001)	Klatt 2020 [54]
	**Yogic breathing technique** (40 days)	pre < post (*p* = 0.015)	Divya 2021 [56]
	VR relaxation (1 session)	Statistical analysis was not performed.	Nijland 2021 [64]
3. UWES (work engagement)	**Mindfulness-based movement** (8 weeks)	(1) Total: pre < post (*p* = 0.00001); (2) Vigor: pre < post (*p* = 0.00001); (3) Absorption: pre < post (*p* = 0.00001); (4) Dedication: pre < post (*p* = 0.00001)	Klatt 2020 [54]
4. Brief Resilience Scale (resilience)	Wellness program (NR)	NR	Coffey 2021 [55]
Outcomes related to well-being			
1. WEMWBS (well-being)	**Mindfulness breathing meditation** vs. no intervention (4 weeks)	EG < CG (*p* = 0.013)	Ibrahim 2022 [70]
	Well-being program (4–6 months)	NS (*p* = 0.34)	So 2021 [67]

Abbreviations. CAMS, Cognitive and Affective Mindfulness Scale; CBT, cognitive-behavioral therapy; CCT, non-randomized controlled clinical trial; CDRS, Connor–Davidson Resilience Scale; CG, control group; DAS, Templer Death Anxiety Scale; DASS, Depression, Anxiety and Stress Scale; DTS, Davidson Trauma Scale; EG, experimental group; ENSS, Expanded Nursing Stress Scale; ISI, Insomnia Severity Index; MAIA, Multidimensional Assessment of Interoceptive Awareness; MBI–HSS, Maslach Burnout Inventory–Human Services Survey; MBI, Maslach Burnout Inventory; MBSR, mindfulness-based stress reduction; NR, not reported; PFI, Stanford Professional Fulfillment Index; PHQ, Patient Health Questionnaire; ProQOL, Professional Quality of Life; PSQI, Pittsburgh sleep quality index; PSS, Perceived Stress Scale; RCT, randomized controlled clinical trial; SAS, Self–rating Anxiety Scale; SCS, Self-Compassion Scale; SDS, Self–rating Depression Scale; STAI, State–Trait Anxiety Inventory; SWLS, Satisfaction With Life Scale; UWES, Utrecht Work Engagement Scale; VR, virtual reality; WEMWBS, Warwick–Edinburgh Mental Well-being Scale. Note. If it was associated with a statistically significant benefit compared to the control group, the intervention was highlighted in bold.

## Data Availability

The data presented in this study are available in the article and Appendix A.

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
