# Peer review of "Systematic Review of Mind–Body Modalities to Manage the Mental Health of Healthcare Workers during the COVID-19 Era"

_healthcare, 2022, doi:10.3390/healthcare10061027_

Round 1

Reviewer 1 Report

The introductory part is appropriately written and clearly indicates the reasons for conducting this research. The authors stated that the "review aims to investigate the role of MBM's in managing the mental health of HCWs healthcare workers during the COVID-19 pandemic".

The chapter Materials and Methods presents to the reader in a clear and concise way the methodology used in the paper.

The Results chapter in fact shows the method of selecting the studies included in the analysis, and Tables 1, 2 and 3 show the characteristics of the included studies and the methodological qualities of the included studies. In my opinion, the above mentioned would be more appropriate to be presented in the Materials and Methods chapter. Furthermore, Table 1, although informative, is very extensive and difficult to follow. In my opinion, it would be more appropriate to present those data in text format. Likewise, the contents of Tables 2 and 3 would also be better to be presented textually.

In my opinion, the real results of this review are presented only in Table 4 and Figure 2 as they show the effects of individual mind-body modality method on mental health management in health professionals during the COVID-19 pandemic. In the text part of subchapter 3.4. Main Results, Figures 3a, 3b and 3c are mentioned. However, there are only two figures in the manuscript. Therefore, the text should be corrected to Figure 2a, 2b and 2c. Furthermore, the textual part is incorrectly related to certain parts of Figure 2. For example, in the text written in rows 294-296 is stated that "mindfulness-based intervention showed borderline significance compared to the psychoeducational intervention (SMD, -0.18; 95% CI, -0.36 to -0.00) (Figure 3a)." However, in the SMD column of the Figure 2. such SMD of -0,18 is present only in the Figure 2c. I suggest to the authors to thoroughly check the data in the text and to adjust them with those present in the Figure 2.

Chapter Discussion in subchapter 4.1. summarizes the Main Findings from subchapter 3.4. Main Results. In the next subchapter 4.2. Clinical Implications the authors concisely but reasonably discuss the possibility of introducing certain mind-body modalities methods into the clinical environment in the context of COVID-19. Subchapter 4.3. Limitations and Suggestions for Further Studies as well as Chapter 5. Conclusion are written correctly.

Author Response

  • Response to Comments from Reviewer 1

Comment 1:

The introductory part is appropriately written and clearly indicates the reasons for conducting this research. The authors stated that the "review aims to investigate the role of MBM's in managing the mental health of HCWs healthcare workers during the COVID-19 pandemic".  

Response:              

Thank you very much for taking your valuable time to review this manuscript. We have no doubt that the reviewer’s comments will help to further improve the quality of this manuscript.

Comment 2:

The chapter Materials and Methods presents to the reader in a clear and concise way the methodology used in the paper. 

Response:              

Thank you for your comment.

Comment 3:

The Results chapter in fact shows the method of selecting the studies included in the analysis, and Tables 1, 2 and 3 show the characteristics of the included studies and the methodological qualities of the included studies. In my opinion, the above mentioned would be more appropriate to be presented in the Materials and Methods chapter. Furthermore, Table 1, although informative, is very extensive and difficult to follow. In my opinion, it would be more appropriate to present those data in text format. Likewise, the contents of Tables 2 and 3 would also be better to be presented textually.

Response:              

Thank you for your comment. We agree that, as the reviewer points out, Tables 1-3 interfere with the readability of this manuscript.

In the case of Table 1, since it contains important information of this review, it was preserved in the revised manuscript, but the contents were summarized and the table has been converted horizontally to increase its readability. On the other hand, in the case of Tables 2 and 3, they were integrated and separated into supplementary file 3.

(Please refer Table 1 and Supplementary file 3)

Comment 4:

In my opinion, the real results of this review are presented only in Table 4 and Figure 2 as they show the effects of individual mind-body modality method on mental health management in health professionals during the COVID-19 pandemic. In the text part of subchapter 3.4. Main Results, Figures 3a, 3b and 3c are mentioned. However, there are only two figures in the manuscript. Therefore, the text should be corrected to Figure 2a, 2b and 2c. Furthermore, the textual part is incorrectly related to certain parts of Figure 2. For example, in the text written in rows 294-296 is stated that "mindfulness-based intervention showed borderline significance compared to the psychoeducational intervention (SMD, -0.18; 95% CI, -0.36 to -0.00) (Figure 3a)." However, in the SMD column of the Figure 2. such SMD of -0,18 is present only in the Figure 2c. I suggest to the authors to thoroughly check the data in the text and to adjust them with those present in the Figure 2.

Response:              

Thank you for your comment. We are very sorry for the error. What was previously described as ‘Figure 3’ is Figure 2, not Figure 3. This error has been corrected in this revised version. Also, we have corrected the order in the image (2(a) to 2(c)), and thoroughly double-checked the data in the text.

(Please refer page 10, red words)

Comment 5:

Chapter Discussion in subchapter 4.1. summarizes the Main Findings from subchapter 3.4. Main Results. In the next subchapter 4.2. Clinical Implications the authors concisely but reasonably discuss the possibility of introducing certain mind-body modalities methods into the clinical environment in the context of COVID-19. Subchapter 4.3. Limitations and Suggestions for Further Studies as well as Chapter 5. Conclusion are written correctly..

Response:              

Thank you for your comment.

Reviewer 2 Report

The present review examined the role of engaging in wellness activities like meditation, yoga, mindfulness exercises as well as interventions based on cognitive behavioral therapy in improving mental health outcomes in healthcare workers during the Covid-19 pandemic. The authors have done a good work in summarizing selected studies and succinctly presenting results. However, there are a number of concerns that should be addressed before the paper is considered for publication.  

In the introduction section a clear definition of what authors intend with “mind-body modality” is lacking. This is necessary considering that authors lump together quite different type of studies that focus both in wellness exercises/activities like meditation, yoga, mindfulness exercises as well as more structured interventions based on cognitive-behavioral therapy principles. Considering such wide range of studies included in the review, care should be taken that differences between the type of activity/exercise and/or intervention are clearly made, especially in light of the very different principles they entail.

I suggest authors to substantially review their introduction section (in view of the comment above) and especially avoid making sweeping statements such as “Their mental health can lead to burnout, worsening attitudes toward patient safety, and hindering the efficient and safe use of medical resources when they are important” in lines 35-35. What they perhaps mean in this case  is mental health problems or difficulties.

In the Methods section, authors lump together under the subheading “Types of intervention” a plethora of activities from mere breathing exercises to more structured interventions (i.e. cognitive-behavioral) are lumped together. I wander whether all these types of activities deserve to be named as “interventions”?

A related point, throughout the text it seems that terms “mind-body modality”, intervention, therapy are used interchangeably. I think clarity should be made as to these definitions and there should be more consistency in the terminology.

In Table 3, it would be more readable if instead of headings Q1, Q2, Q3 and so on brief etiquettes could be provided.

Author repeatedly make reference to a Figure 3 in the text which is no where to be found.

In the Main Findings section, lines 352-353 the phrase “… the effects of HCWs (healthcare workers) on mental health may differ according to individual MBMs.” is incomprehensible.

In the Conclusion section, lines 427-429 the following phrase seems contradictory: “In addition, some MBMs had significant positive effects on psychological trauma, burnout, insomnia, anxiety, depression, self-compassion, mindfulness, QOL, resilience, and well-being, but not psychological trauma and self-efficacy, compared with the controls or baseline.” Please reframe.

Author Response

  • Response to Comments from Reviewer 2

Comment 1:

The present review examined the role of engaging in wellness activities like meditation, yoga, mindfulness exercises as well as interventions based on cognitive behavioral therapy in improving mental health outcomes in healthcare workers during the Covid-19 pandemic. The authors have done a good work in summarizing selected studies and succinctly presenting results. However, there are a number of concerns that should be addressed before the paper is considered for publication. 

Response:              

Thank you very much for taking your valuable time to review this manuscript. We have no doubt that the reviewer’s comments will help to further improve the quality of this manuscript. Specifically, we acknowledge that there were some unclear and ambiguous sentences. We have made every effort to minimize unclear or ambiguous expressions in this revised manuscript.

Comment 2:

In the introduction section a clear definition of what authors intend with “mind-body modality” is lacking. This is necessary considering that authors lump together quite different type of studies that focus both in wellness exercises/activities like meditation, yoga, mindfulness exercises as well as more structured interventions based on cognitive-behavioral therapy principles. Considering such wide range of studies included in the review, care should be taken that differences between the type of activity/exercise and/or intervention are clearly made, especially in light of the very different principles they entail.

Response:              

Thank you for your comment. In the Introduction section of this revised version, we described the definition of mind-body modality.

“Mind–body modality (MBM) can be defined as “a health practice that combines mental focus, controlled breathing, and body movements to help relax the body and mind [6].””

(Please refer page 1, red words)

And the explanations of terms ‘modality’, ‘therapy’, and ‘intervention’ were included in the Comments 4 and 5.

Comment 3:

I suggest authors to substantially review their introduction section (in view of the comment above) and especially avoid making sweeping statements such as “Their mental health can lead to burnout, worsening attitudes toward patient safety, and hindering the efficient and safe use of medical resources when they are important” in lines 35-35. What they perhaps mean in this case  is mental health problems or difficulties.

Response:              

Thank you for your comment. The sentence pointed out by the reviewer has been corrected, and in addition to that, the Introduction part has been revised in general to avoid inclusive and ambiguous statements and expressions.

“Mental health difficulties of HCWs can lead to burnout, worsening attitudes toward patient safety, and hindering the efficient and safe use of medical resources when they are important [4].”

(Please refer pages 1-2, red words)

Comment 4:

In the Methods section, authors lump together under the subheading “Types of intervention” a plethora of activities from mere breathing exercises to more structured interventions (i.e. cognitive-behavioral) are lumped together. I wander whether all these types of activities deserve to be named as “interventions”?

Response:              

Thank you for your comment. The term ‘intervention’ in the “Types of intervention” is derived from “interventional study” among types of clinical studies. That is, any treatment or strategy in a clinical study conducted to investigate its effect on (usually) prospectively recruited participants can be considered an intervention.

The interventions can be quite varied; examples include administration of a drug or vaccine or dietary supplement, performance of a diagnostic or therapeutic procedure, and introduction of an educational tool. Depending on whether the intervention is aimed at preventing the occurrence of a disease (e.g., administration of a vaccine, boiling of water, distribution of condoms or of an educational pamphlet) or at providing relief to or curing patients with a disease (e.g., antiretroviral drugs in HIV-infected persons), a trial may also be referred to as “preventive trial” or “therapeutic trial”.

[reference] Aggarwal R, Ranganathan P. Study designs: Part 4 - Interventional studies. Perspect Clin Res. 2019 Jul-Sep;10(3):137-139.

Therefore, even in the case of mere breathing exercise, there are some cases where this is described as breathing exercise intervention.

[reference] Rodrigues SN, Henriques HR, Henriques MA. Effectiveness of preoperative breathing exercise interventions in patients undergoing cardiac surgery: A systematic review. Rev Port Cardiol (Engl Ed). 2021 Mar;40(3):229-244.

[reference] Bahenský P, Bunc V, Malátová R, Marko D, Grosicki GJ, Schuster J. Impact of a Breathing Intervention on Engagement of Abdominal, Thoracic, and Subclavian Musculature during Exercise, a Randomized Trial. J Clin Med. 2021 Aug 10;10(16):3514.

[reference] Laborde S, Hosang T, Mosley E, Dosseville F. Influence of a 30-Day Slow-Paced Breathing Intervention Compared to Social Media Use on Subjective Sleep Quality and Cardiac Vagal Activity. J Clin Med. 2019 Feb 6;8(2):193.

[reference] Dhawan A, Chopra A, Jain R, Yadav D, Vedamurthachar. Effectiveness of yogic breathing intervention on quality of life of opioid dependent users. Int J Yoga. 2015 Jul-Dec;8(2):144-7.

Nevertheless, if the reviewer still thinks there is a problem with the use of the term intervention, we are willing to look for a better term.

Comment 5:

A related point, throughout the text it seems that terms “mind-body modality”, intervention, therapy are used interchangeably. I think clarity should be made as to these definitions and there should be more consistency in the terminology.

Response:              

Thank you for your comment.

In general, although some mind-body modalities can be provided as medical treatments in clinical settings, the terms ‘mind-body modalities’ or ‘mind-body practices’ are preferred over the term ‘mind-body therapy’, as they can also be trained as self-care skills of individuals or trained in a non-medical institution.

However, in the case of some mind-body modalities, the term ‘therapy’ can be attached, when there is a specialized therapist. Examples include music therapy (by music therapist) and aromatherapy (by aromatherapist). Therefore, the term ‘music therapy’ was included in the manuscript.

Irrespective of ‘modality’ or ‘therapy’, both can be regarded as ‘interventions’ in the interventional study. That is, mind-body modalities can be applied as ‘interventions’ to the participants by the intention of the researcher in clinical study settings.

Comment 6:

In Table 3, it would be more readable if instead of headings Q1, Q2, Q3 and so on brief etiquettes could be provided.

Response:              

Thank you for your comment. Tables 2 and 3 were integrated and separated into supplementary file 3. Also, key evaluation indicators have been added to each question to improve readability.

(Please refer Supplementary file 3)

Comment 7:

Author repeatedly make reference to a Figure 3 in the text which is no where to be found.

Response:              

We are very sorry for the error. What was previously described as ‘Figure 3’ is Figure 2, not Figure 3. This error has been corrected in this revised version.

(Please refer page 10, red words)

Comment 8:

In the Main Findings section, lines 352-353 the phrase “… the effects of HCWs (healthcare workers) on mental health may differ according to individual MBMs.” is incomprehensible.

Response:              

Thank you for your comment. The sentence pointed out by the reviewer has been corrected, and in addition to that, the Discussion part has been revised in general to avoid inclusive and ambiguous statements and expressions.

“In other words, the effects of MBMs on mental health of HCWs may differ according to individual MBMs.”

(Please refer pages 17-18, red words)

Comment 9:

In the Conclusion section, lines 427-429 the following phrase seems contradictory: “In addition, some MBMs had significant positive effects on psychological trauma, burnout, insomnia, anxiety, depression, self-compassion, mindfulness, QOL, resilience, and well-being, but not psychological trauma and self-efficacy, compared with the controls or baseline.” Please reframe.

Response:              

Thank you for your comment. The sentence pointed out by the reviewer has been corrected.

“In addition, some MBMs had significant positive effects on psychological trauma, burnout, insomnia, anxiety, depression, self-compassion, mindfulness, QOL, resilience, and well-being, but not psychological trauma and self-efficacy of HCWs.”

(Please refer page 18, red words)

Round 2

Reviewer 2 Report

No further comments.